



# Ice crystal c-axis orientation and mean grain size measurements from the Dome Summit South ice core, Law Dome, East Antarctica

A. Treverrow[1], J. Li[2], and T. H. Jacka[1]

[1]Antarctic Climate and Ecosystems Cooperative Research Centre, University of Tasmania, Hobart, Australia, 7004
[2]SGT Inc., NASA Goddard Space Flight Center, Greenbelt, MD, USA

*Correspondence to:* Adam Treverrow (adam.treverrow@utas.edu.au)

**Abstract.** We present measurements of crystal $c$-axis orientations and mean grain area from the Dome Summit South (DSS) ice core drilled on Law Dome, East Antarctica. These data are from 185 individual thin sections obtained between a depth of $117\,\mathrm{m}$ below the surface and the bottom of the DSS core at a depth of $1196\,\mathrm{m}$. The median number of $c$-axis orientations recorded in each thin section was 100, with values ranging from 5 through to 111 orientations. The data from all 185 thin sections are provided in a single comma separated value (csv) formatted file which contains the $c$-axis orientations in polar coordinates, depth information for each core section from which the data were obtained, the mean grain area calculated for each thin section and other data related to the drilling site. The data set is also available as a MATLAB™ structure array. Additionally, the $c$-axis orientation data from each thin of the 185 thin sections are summarised graphically in figures containing a Schmidt diagram, histogram of $c$-axis colatitudes and rose plot of $c$-axis azimuths. All of these data are referenced by doi:10.4225/15/5669050CC1B3B and are available free of charge at https://data.antarctica.gov.au.

## 1 Introduction

The greatest source of uncertainty in forecasts of sea level rise during the 21st century is the contribution from the Antarctic and Greenland ice sheets (e.g. Willis and Church, 2012; Gregory et al., 2012). The poor constraints on predictions of grounded ice discharge are related to the currently inadequate description of ice dynamic processes and boundary conditions in the numerical models used to simulate ice sheet evolution (e.g. Alley and Joughin, 2012; Gregory et al., 2013; Vaughan et al., 2013; Carson et al., 2014). One of the key components of all numerical ice sheet models is the relationship which governs ice flow rates as a function of temperature and the stresses driving the deformation. The primary consideration in the development of a numerical flow relation for ice sheet modelling is to provide a realistic description of ice rheology that does not significantly decrease the computational efficiency of the model.

Typically, ice is modelled as a very slow flowing, non-linear viscoelastic fluid where the effective viscosity is highly temperature dependent, varying by $\sim 3$ orders of magnitude over the in situ temperature range. Despite being modelled as a high viscosity fluid, the polar ice sheets are massive polycrystalline aggregates of solid ice, where the largest linear dimension of individual grains is in the order of several mm to cm (e.g. Faria et al., 2014a; Ng and Jacka, 2014). Furthermore, ice flow rates



are significantly influenced by the microstructural evolution, which occurs during deformation (e.g. Budd and Jacka, 1989; Cuffey and Paterson, 2010).

Under terrestrial conditions ice exists in the hexagonal Ih phase and individual crystals possess a high level of plastic anisotropy because their dominant mode of deformation is slip on crystallographic basal planes (e.g. Duval et al., 1983; Schulson and Duval, 2009; Cuffey and Paterson, 2010). Within a polycrystalline aggregate, the spatial orientation of the basal planes of an individual crystal is defined by the orientation of its crystallographic $c$-axis, which is also the axis of hexagonal symmetry in an ice crystal and the normal to the basal planes. During the high-strain deformation, typical of the polar ice sheets, patterns of preferred crystal $c$-axis orientations (often referred to as a crystal orientation fabrics) evolve to accommodate crystallographic slip on basal planes. This leads to the development of polycrystalline anisotropy and an associated reduction in the ice viscosity (e.g. Budd and Jacka, 1989; Cuffey and Paterson, 2010). A physically accurate parameterisation of these effects is fundamental to improving the predictive capability of ice sheet models, in order to i) more accurately predict ice sheet contributions to global sea level, and ii) reduce uncertainty in the depth-age relationships used to constrain ice core palaeoclimate records.

Various numerical flow relations for ice, where the effect of $c$-axis orientations on the flow properties is explicitly included as a rheological variable, have been proposed (e.g. Lile, 1978; van der Veen and Whillans, 1994; Azuma and Goto-Azuma, 1996; Svendsen and Hutter, 1996; Thorsteinsson, 2002; Gillet-Chaulet et al., 2005; Placidi et al., 2010). In such flow relations the anisotropic viscosity of a polycrystalline aggregate is derived from the orientation relationship between the grain $c$-axes and the stress configuration, or some parameterisation of these orientation effects. In general such flow relations are not suited to implementation within models used to simulate the large-scale evolution of the polar ice sheets, being either too numerically complex or lacking the ability to accurately describe anisotropic flow effects (e.g. Treverrow et al., 2015). Accordingly, the ongoing value of such physically-motivated flow relations lies in their role as tools to understand intra- and intercrystalline microdeformation, recovery and recrystallisation processes (e.g. Montagnat et al., 2014b; Faria et al., 2014b). In turn, these models can inform the development of parameterisations of key rheological variables, which are necessary for the specification of numerically simpler ice flow relations.

The continued development and validation of physically accurate flow relations for ice sheet modelling requires observations of ice microstructures from deep drilled ice cores, including the patterns of preferred crystal $c$-axis orientations within an ice mass. Such data can be obtained from the analysis of thin section samples. These measurements are made routinely in ice core and laboratory deformation studies investigating the links between the microstructure of ice and the large-scale dynamics of ice sheets (e.g. Gow and Williamson, 1976; Russell-Head and Budd, 1979; Gao and Jacka, 1987; Pimienta et al., 1987; Budd and Jacka, 1989; Tison et al., 1994; Li et al., 2000; Durand et al., 2007; Gow and Meese, 2007; Treverrow et al., 2012; Montagnat et al., 2014a).

Here we present ice crystallographic $c$-axis orientation and grain size data from the Dome Summit South (DSS) ice core drilled 4.7 km SSW of the summit of Law Dome, East Antarctica (66.770°S, 112.807°E, Table 1,Li, 1995; Morgan et al., 1997; Li et al., 1998; Morgan et al., 1998). This 1195.9 m ice core was drilled by the Australian Antarctic Division during the austral summers of 1987-88 to 1992-93. Law Dome (Figure 1) is a coastal ice cap $\sim 200$ km in diameter with a summit




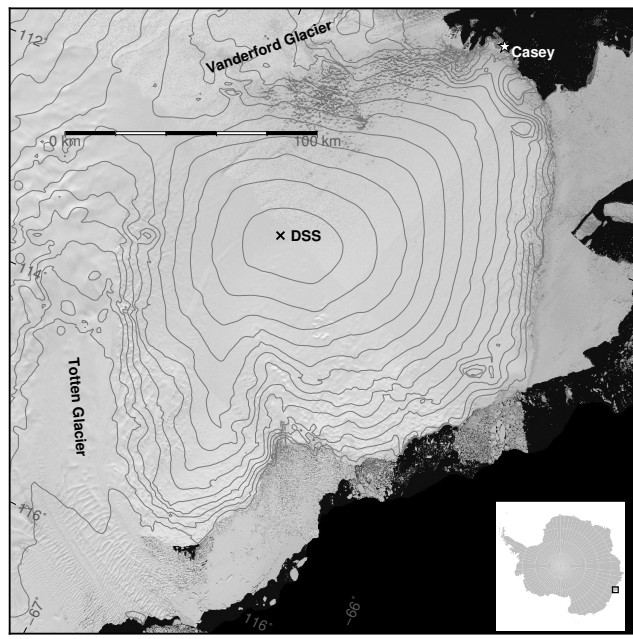

**Figure 1.** Location of the drilling site for the $1196\,\mathrm{m}$ Dome Summit South (DSS) ice on Law Dome, East Antarctica. The drill site is $4.7\,\mathrm{km}$ SSW from the dome summit. The background image is from the Landsat Image Mosaic of Antarctica (Bindschadler et al., 2008) and $100\,\mathrm{m}$ elevation contours from Bamber et al. (2009).

elevation of $1370\,\mathrm{m}$. Grounded ice at Law Dome extends to lower latitudes than any other region of the Antarctic Ice Sheet, with the exception of the northern Antarctic Peninsula. As such the local climate has a strong maritime influence and the summit region of Law Dome experiences a high rate of annual snow accumulation and relatively low wind speeds (Bromwich, 1988; Curran et al., 1998). The ice dynamics of Law Dome are relatively independent of the adjoining East Antarctic Ice Sheet

5 as it is isolated from the Aurora sub-glacial basin by the Vanderford trench (Pfitzner, 1980; Roberts et al., 2011).

A primary consideration in the selection of the DSS drilling site was identification of a location where the rates of ice deformation, and those upstream, were insufficient to significantly disturb the chronology of the annually accumulated ice layers. Ice penetrating radar was used to assist site selection by identifying locations where the regional bedrock topography was least likely to have created flow-induced folding or other discontinuities in the ice laminae (Hamley et al., 1986;

10 Etheridge, 1990; Morgan et al., 1997). The well-preserved layering at the DSS site, combined with the maritime influence on the orographically-driven high accumulation rates on Law Dome has made the DSS ice core a valuable resource for generating mid- and high-latitude palaeoclimate proxy records from the Holocene and Last Glacial Maximum (LGM; van Ommen and Morgan, 2010; Plummer et al., 2012; Vance et al., 2013; Roberts et al., 2015; Vance et al., 2015).





**Table 1.** DSS borehole site and ice core information (Morgan et al., 1997, 1998)

| | |
|---|---|
| Latitude | $66.770°$S |
| Longitude | $112.807°$E |
| Surface elevation | $1370$m |
| Ice thickness (RES) | $1220 \pm 20$m |
| Borehole depth | $1195.9$m |
| Borehole ice equivalent depth | $1174.07$m |
| Mean annual surface temperature | $-21.8\,°$C |
| Bottom of borehole temperature | $-6.9\,°$C |
| Annual ice accumulation | $0.69\,\mathrm{m\,a^{-1}}$ |
| Ice surface velocity and bearing | $(2.04 \pm 0.11)\,\mathrm{m\,a^{-1}}$ at $225° \pm 3°$ |
| Surface GPS strain rate (parallel to surface flow direction) | $(3.22 \pm 0.016) \times 10^{-4}\,\mathrm{a^{-1}}$ |
| Surface GPS strain rate (normal to surface flow direction) | $(4.50 \pm 0.027) \times 10^{-4}\,\mathrm{a^{-1}}$ |

## 2 Methods

The upper $96$m of the DSS core was obtained with a $270$mm diameter thermal drill during the 1987-1988 austral summer
and the borehole was cased to a depth of $82$m. In the following season a drill shelter was constructed over the borehole and
thermal drilling continued to $117$m depth using a smaller $120$mm drill. Over the 1989-1990 and 1990-1991 field seasons the
electromechanical drill used to recover the main DSS ice core was assembled and commissioned (Morgan et al., 1997). This
drill was based on a modified Danish ISTUK design (Gundestrup et al., 1984) and produced $100$mm diameter core sections.
Coring to a depth of $553.9$m was completed during the 1991-1992 field season and in the following season drilling continued
to the final depth of $1195.6$m. Coring was halted when the cutting head of the drill was damaged through contact with a rock,
which was not recovered in the core. This final core section was found to contain other, small rock fragments. The estimated
total ice thickness of $1220 \pm 20$m at the DSS site was determined using ice penetrating radar (Morgan et al., 1997). Based on
this value the bedrock is expected to be within $5$m to $45$m of the bottom of the borehole. The DSS core was recovered in 1261
sections with a mean length of $0.949$m and standard deviation, $\sigma = 0.092$m.

### 2.1 Thin section preparation

Ice crystal $c$-axis orientation and grain size measurements were made on horizontal thin sections obtained from 185 of the
1261 individual ice core sections that make up the DSS core. These samples were obtained at intervals of $\sim 5$ to $6$m between
the depths of $117.1$m and $1195.9$m. The thin sections were prepared during drilling operations in order to minimise the time





between core recovery and analyses. Typically, the interval between drilling and thin section preparation was $< 1$ to $16\,\mathrm{h}$, with a mean of $\sim 3\,\mathrm{h}$ (Li et al., 1998). This procedure reduced the potential for any post-drilling microstructural evolution to adversely influence the analyses. Furthermore, all sample preparation and analyses were conducted in a sub-surface laboratory, excavated adjacent to the drilling pit. Since the temperature in the laboratory was approximately equivalent to the mean annual surface

temperature of $-21.8\,°\mathrm{C}$ all analyses were conducted at temperatures below the in situ values. Working at these low temperatures reduced the potential for post-drilling microstructural modification. Conducting the microstructural observations at the drilling site also eliminated the risk of damaging the samples during long-term storage and/or transportation to laboratories outside Antarctica.

    The $\sim 10\,\mathrm{mm}$ thick horizontal sections were cut perpendicular to the long axis of the ice core. These samples were sanded to

a smooth finish using wet-and-dry type abrasive paper mounted on a flat board before being thermally bonded to glass slides. Following mounting, the thickness of the samples was reduced to between $0.4\,\mathrm{mm}$ and $0.7\,\mathrm{mm}$ using a microtome. Thicknesses towards the lower end of this range were necessary to clearly resolve individual grains in those samples with a smaller mean grain size. Two different microtomes were used to reduce the sample thickness. An electromechanical microtome, based on a modified wood-working plane, was better suited to rapidly reducing the section thickness while a manually operated sledge

microtome provided superior control when making the final fine adjustments to the section thickness. The sledge microtome was also found to be less likely to induce fracturing when working with the extremely brittle ice encountered at depths from $552\,\mathrm{m}$ to $1190\,\mathrm{m}$.

## 2.2   Crystallographic c-axis measurements

The crystal $c$-axis orientations were measured manually using a universal (Rigsby) stage following the standard techniques

described by Langway (1958). The instrument used was modified to provide digital orientation data for each grain examined (Morgan et al., 1984). These output data are corrected for differences in the refractive indices of ice and air using the values of Langway (1958).

    As outlined by Langway (1958), several potential sources of error can influence the orientations determined using a universal stage. These include: i) uncertainty in precisely locating the extinction position for high angle orientations, ii) systematic

operator bias due to not correctly aligning the viewing direction (line of sight) with the crystal $c$-axis being examined, and iii) instrument backlash due to inherent and finite non-zero tolerances that influence the reproducibility of measurements. Langway (1958) estimates that the latter may contribute to $1°$ to $2°$ of measurement error and that the combined maximum error from all sources will typically be $< 5°$, particularly when measurements are made with consistently high levels of care. Because the universal stage used to measure the DSS ice core crystal orientations employs sensors to determine the position of the

instrument axes, incorrect reading of the A1 and A2 axis scales is eliminated as a source of error in these data. An upper limit of the possible angular error in azimuth or colatitude data is $\pm 5°$.

    Due to the time consuming procedure required to manually determine $c$-axis orientations using the universal stage in a challenging work environment, a maximum of $\sim 100$ orientations were recorded from each thin section, including those fine-grained samples where the total number of grains, $N$, within the thin section was $\gg 100$. Analyses were completed at the rate



of $\sim 100$ $c$-axis orientations per 3 h. For those thin sections containing larger grains with $N \ll 100$, $c$-axis orientations were recorded for all identifiable grains. For each of the 185 DSS thin sections, the number of measured $c$-axis orientations was $5 < N < 111$, with a median of $N = 100$.

During the 1990s when the DSS ice core was drilled, the measurement of ice crystal $c$-axis orientations using a universal stage, as used in this study, was considered state-of-the-art. In the time since these measurements were made considerable technological advances in the instruments available to measure crystal orientations have occurred (e.g. Yun and Azuma, 1999; Wilen et al., 2003; Wilson et al., 2007; Wilson and Peternell, 2011). Modern instruments provide several benefits over the universal stage including: i) higher resolution, enabling small-scale differences in crystallographic orientations within individual grains to be detected, ii) the ability spatially map $c$-axis orientations, iii) automated operation and significantly higher speed of determining orientations and iv) lower levels of uncertainty in both $c$-axis azimuth and colatitude. Since analytical techniques for both microstructural and chemical analyses of ice cores are destructive, only a small proportion of the original ice core cross section remains over the full length of the core. This remainder is insufficient to allow a detailed reanalysis of $c$-axis orientations and grain size using a modern instrument. Additionally, the remaining core material is prioritised for chemical reanalyses, should any be required.

## 2.3 Crystal size measurements

The mean grain size was determined from polaroid photographs of the thin sections. A purpose built stand was used to position the thin sections between orthogonal plane polarising filters. This allowed individual grains to be visually distinguished by their orientation-dependent birefringence colours. A polaroid camera, mounted directly to the stand, captured high quality colour images at 1:1 scale. A transparent cover plate, etched with a 10 mm square grid was placed over the samples to superimpose a grid of the same dimensions onto the polaroid images.

The mean grain area was calculated from the number of grains contained within a specified area of the thin section. The region of interest was determined by placing a sheet of low opacity tracing paper over the polaroid image; its transparency allowed an irregularly shaped region to be traced along grain boundaries and the number of grains within this region to be counted. A digitizing tablet was used to accurately determine the area of the irregularly shaped region marked on the tracing paper. Typically these areas for grain size analysis varied between 880 and 2400 mm$^2$ (Li, 1995). The area, in conjunction with the total number of grains counted within the traced region, was used to calculate an arithmetic mean estimate of cross sectional grain area. Figures incorporating the depth profile of mean grain size at the DSS site have been presented previously, without the data being made publicly available (e.g. Morgan et al., 1997; Li et al., 1998).

Uncertainty in the mean grain area estimates may originate from errors counting the number of grains within the traced region or determining the area of the traced region when using a digitizing tablet. For digitizing tablets, instrument related planimetric position errors are typically $\ll 1$ mm and are negligible in comparison with any operator error. Consequently, the uncertainty in the calculated mean grain areas is strongly related to both operator skill and consistency.

To determine an upper limit on the uncertainty in the calculated mean area we assume the maximum error in the position of any section of the traced boundary enclosing the counted grains to be $\sim \pm 1$ mm. Applying this level of uncertainty to the entire





perimeter of the traced region results in an area uncertainty of $\sim \pm 10\%$. In most cases this upper limit on uncertainty will be an overestimate of the true uncertainty; however, precise estimates of uncertainty were not made at the time of measurement so we are unable to calculate the specific error for individual mean grain areas.

Some further general comments on the methods used to specify the grain size of polycrystalline materials are warranted. While thin section analysis is the only practical means of routinely estimating grain area, it only provides a two dimensional estimate of a volumetric object. A further consequence of thin section based analyses is that all methods used to determine grain area from thin sections underestimate the actual grain dimensions because the sectioning plane almost never intersects each grain across its plane of maximum cross-sectional area (e.g. Feltham, 1957; Gow, 1969). A variety of techniques for determining the grain size of polycrystalline materials from thin sections, which also take this bias into account exist.

In the methods described by Krumbein (1935); Pickering (1976); Baker (1982); Jones and Chew (1983) grain size is estimated from the mean length of the maximum linear intercept through a grain, which requires assumptions to be made regarding the grain shape and distribution of grain sizes. Stephenson (1967) and Gow (1969) calculated mean grain areas based on the measurement of minimum and maximum axial dimensions of individual grains. To further reduce the effect of sectioning on underestimating the three-dimensional distribution of grain size, Gow (1969) restricted measurement to the 50 largest grains identified within a thin section; however, the results obtained using this technique are sensitive to number of grains within the section.

Methods of grain size analysis that require measurement of linear intercepts through grains (e.g. Krumbein, 1935; Stephenson, 1967; Gow, 1969; Baker, 1982) are time consuming and were considered incompatible with the program of obtaining crystal $c$-axis orientation and size measurements during drilling and field analysis of the DSS ice core. As noted by Jacka (1984), the mean linear intercept methods of grain size estimation, and in particular those assumptions that attempt to correct for sectioning effects, may be inappropriate when the actual distributions of grain size and shape are either unknown or variable. In such cases Jacka (1984) suggests that the arithmetic mean grain area, based on the crystal count per unit area, is a superior estimate of grain size. For polycrystalline metals Feltham (1957) has demonstrated that the three-dimensional distribution of grain size can be represented by the planar distribution obtained from thin sections. Provided that grain shape is not correlated with the crystallographic $c$-axis orientation this result is directly applicable to other polycrystalline materials, including ice. Therefore the mean grain area from the count per unit area for a particular sample can be considered representative of the volumetric grain size. While the mean grain areas presented here will underestimate the corresponding three-dimensional grain size due to the aforementioned sectioning effects, the data remain valuable as they are indicative of the mean grain size at a specific depth and represent the evolution in grain size as a function of depth.

# 3   Data

Measurements of $c$-axis orientations and mean grain area were made on horizontal thin sections obtained from the upper end of each core section. The reported depth for each thin section is that from the ice sheet surface to the top of core section. In the upper part of an ice sheet the observed ice density increases with depth as snow undergoes a transition to firn and then glacial



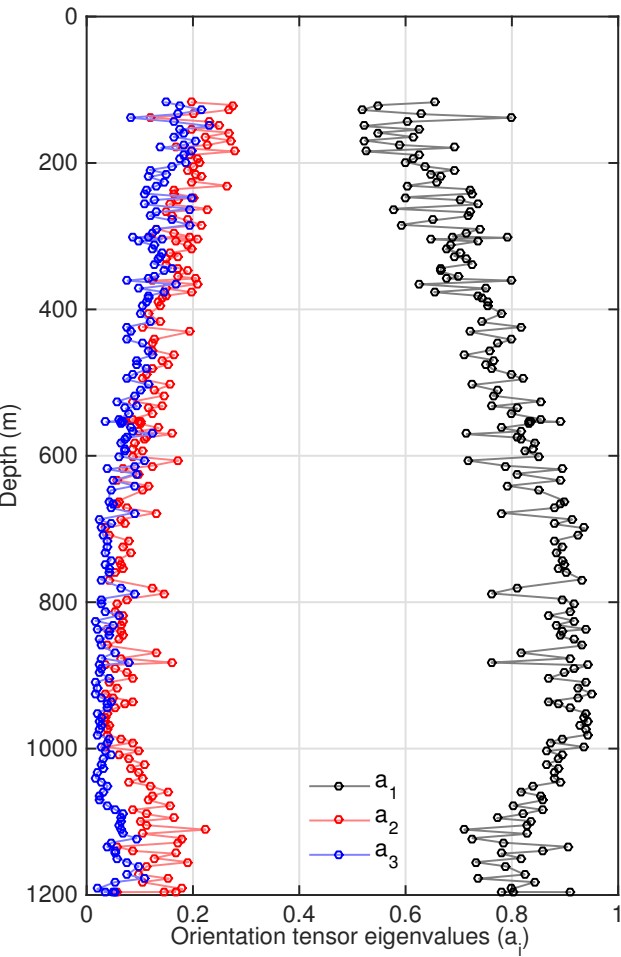

**Figure 2.** Variation in the eigenvalues, $a_i$, of the second-order orientation tensor, $\Lambda$ as a function of depth. See text for details of $a_i$ (after Treverrow et al., 2015).



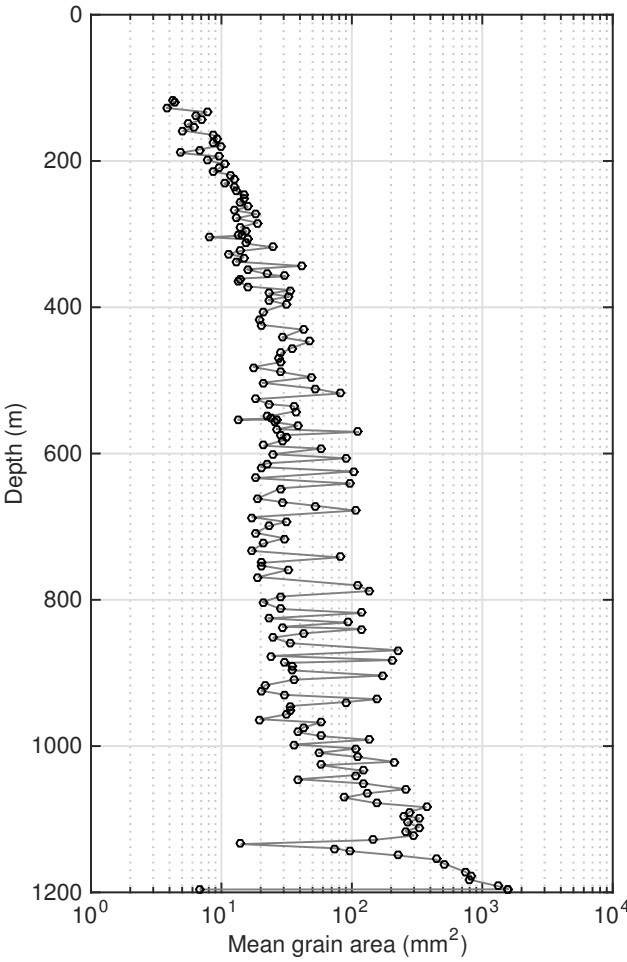

**Figure 3.** Variation in the DSS ice core mean grain area with depth. See text for details of the mean grain area calculation. All values were determined from horizontal thin sections (after Li et al., 1998).



**Table 2.** Description of the data fields in each row of the csv formatted DSS ice core $c$-axis orientation and mean grain area data file, `DSS_fabric_data.csv`

| | |
|---|---|
| name | identifier of the DSS ice core section |
| longitude | DSS ice core site longitude (decimal degrees) |
| latitude | DSS ice core site latitude (decimal degrees) |
| number_of_c_axes | the number of $c$-axes in the section identified by 'name' |
| depth_actual | the actual depth measured downwards from the ice sheet surface to the top of the core section (metres) |
| depth_ice_equivalent | 'depth_actual' converted to an ice equivalent depth (metres) |
| section_orientation | thin section orientation ('vertical' or 'horizontal') |
| mean_horz_grain_area | mean grain area measured from horizontal thin sections (perpendicular to the vertical ice core axis; $\mathrm{mm}^2$) |
| grain_index | the identifier of the data point within the thin section |
| colatitude | $c$-axis orientation colatitude (degrees) |
| azimuth | $c$-axis orientation azimuth (degrees) |

ice. The very slow flow of glaciers and ice sheets is the result of gravity-induced deformation that is dependent upon several factors, including the ice sheet geometry (ice thickness, surface elevation, bedrock topography) and the ice density (e.g. Cuffey and Paterson, 2010).

In the numerical models used to simulate ice sheet dynamics it is more practical to specify a constant ice density when 5 calculating the magnitude of the stresses driving ice flow. The assumption of a constant density is reasonable for ice sheets where ice thicknesses may be hundreds or thousands of metres; however, it is necessary to convert any data sets used for calibration or validation of the models to an ice equivalent depth scale. Similarly the interpretation and application of ice core chemistry in palaeoclimate studies is simplified by conversion to an ice equivalent depth scale. This is particularly convenient



**Table 3.** Format of the MATLAB™ R2015b structure array, `DSS_fabric_data.mat`, which contains DSS ice core crystallographic $c$-axis orientation and mean grain area data.

| | |
|---|---|
| name | DSS ice core section name |
| c_cartesian | [N x 3] array of Cartesian $c$-axis unit vectors. N is the number of $c$-axes in the section |
| c_polar | [N x 2] array of polar $c$-axis vectors; colatitude and azimuth (degree) |
| depth_actual | the actual depth measured downwards from the ice sheet surface to the top of the core section (metres) |
| depth_ice_equivalent | 'depth_actual' converted to an ice equivalent depth (metres) |
| mean_horz_grain_area | mean grain area measured from horizontal thin sections (perpendicular to the vertical ice core axis; $\mathrm{mm}^2$) |
| orientation_tensor_a | eigenvalues, **a**, of the 2nd order orientation tensor |
| orientation_tensor_V | eigenvectors, **V**, of the 2nd order orientation tensor |

when synchronising multiple ice core records from different geographical locations. The crystal $c$-axis orientation and mean grain area data are reported here on both actual and ice equivalent depth scales. At an actual depth $\chi$, the ice equivalent depth

$$d(\chi) = \int\limits_0^{\chi} \frac{\rho(\chi')}{\rho_{ice}} \mathrm{d}\chi', \tag{1}$$

where $\rho(\chi')$ is the firn/ice density profile and $\rho_{ice} = 917 \, \mathrm{kg\,m^{-3}}$ is the maximum ice density (Roberts et al., 2015). The depth-

5    varying $\rho(\chi')$ is obtained from an empirical fit to density measurements from the DSS core (van Ommen et al., 1999). The firn to ice transition marks the depth where the interconnected pore space between individual grains is closed off forming discrete bubbles. At DSS this transition occurs at $\sim 40 \, \mathrm{m}$ depth. Below this depth density variations are negligible and the ice equivalent depth offset from the actual depth has a constant value of $-21.53 \, \mathrm{m}$ (van Ommen et al., 2004).

### 3.1 Crystal orientation and mean grain size data products

10    The combined crystal $c$-axis orientation and mean grain size data are provided in two formats.



1. A single comma separated value (csv) file containing $c$-axis orientation information for all grains in each of the 185 thin sections analysed. Each row in the file contains $c$-axis orientation data for an individual grain expressed in polar coordinates (azimuth and colatitude), the mean grain size calculated for the parent thin section and other information relevant to the measurements. See Table 2 for a full description of the fields in the dataset.

2. A MATLAB™ structure array containing $c$-axis vectors expressed in both polar and Cartesian coordinates (MATLAB and Statistics Toolbox Release 2015b, The MathWorks Inc., Natick, Massachusetts, United States). Other data included in the structure array are described in Table 3. Some derived quantities are also presented in the structure array. These include the eigenvalues $a_i$ $(i = 1, 2, 3)$ of the second-order $c$-axis orientation tensor, $\mathbf{\Lambda}$ and its corresponding eigenvectors $\mathbf{v}$.

The pattern of $c$-axis orientations can be expressed by the eigenvalues, $a_i$, of the $c$-axis orientation tensor, $\mathbf{\Lambda}$ (Fig. 2, Woodcock, 1977). Calculation of the orientation tensor requires transformation of the $c$-axis data from polar to Cartesian coordinates. For a population of $N$ $c$-axis vectors the normalised form of $\mathbf{\Lambda}$ is defined in terms of their $N$ outer products:

$$\mathbf{\Lambda} = \frac{\sum\limits_{i=1}^{N} \hat{c}_i \otimes \hat{c}_i}{N}. \tag{2}$$

The eigenvalues are related as $a_1 + a_2 + a_3 = 1$, where by convention $0 \leq a_3 \leq a_2 \leq a_1 \leq 1$. The eigenvalues provide a
statistical representation of the pattern of $c$-axis orientations as each defines the degree of clustering about their corresponding eigenvectors, $\hat{v}_i$,. The distribution of orientations is minimised about the eigenvector $\hat{v}_1$ of the maximum eigenvalue, $a_1$ whilst $\hat{v}_3$ is the direction about which the distribution is largest. The distribution of orientations becomes smaller, i.e. fabrics become stronger or more clustered as $a_1 \rightarrow 1$, whilst for an isotropic (random) distribution of orientations $a_1 = a_2 = a_3 = \frac{1}{3}$. As the area of individual grain orientations was not recorded, volume weighting of the orientation data (e.g. Durand et al., 2006) was
not possible and all orientations contribute equally to $\mathbf{\Lambda}$ (Fig. 2).

The crystallographic $c$-axis and mean crystal area data are summarised as a function of actual depth in Figs. 2 and 3 respectively.

The data available for download also include graphical representations of the $c$-axis orientations. For each thin section a figure containing three subfigures is provided. These include a lower hemisphere Schmidt plot of the $c$-axes, a rose plot of the
distribution of $c$-axis azimuth, $\theta$, and a histogram of the $c$-axis colatitude, $\phi$. The colatitude histograms are annotated with the number of grains, $N$, in the distribution, the actual depth, ice equivalent depth (IED), eigenvalues, $a_i$, of the orientation tensor $\mathbf{\Lambda}$, mean grain area, $G_a$, mean colatitude, $\phi_{\mathrm{mean}}$ and the cone angle containing the first quartile of $c$-axis colatitudes, $\phi_{\frac{1}{4}}$.

## 4   Summary

Observations of ice microstructures from deep drilled polar ice cores play a vital role in the development and validation of ice
flow relations for numerical ice sheet modelling. In particular, measurements of the patterns of ice crystal $c$-axis orientations



and grain size from thin sections, obtained at regular depth intervals along an ice core, provide detailed information on the relationships between large-scale ice sheet dynamics and microstructural evolution.

Measurements of crystal $c$-axis orientations and grain size were obtained from the Dome Summit South (DSS) ice core, drilled at Law Dome, East Antarctica. The data were obtained from 185 horizontal thin sections taken at intervals of approximately 5 to 6 metres between the depths of 117m and 1196m. All $c$-axis orientation measurements were made using a Universal (Rigsby) stage. The number of $c$-axis orientations recorded in each thin section varied according to the grain size and ranged from a minimum of 5 up to 111, with a median value of 100. For each thin section the arithmetic mean grain area was also determined. These data are made available in two formats: i) as a single csv formatted file containing all $c$-axis orientations (in polar coordinates) and mean grain area for each of the 185 thin sections, plus other data relevant to the DSS drilling site and, ii) as a MATLAB™ structure array containing the $c$-axis and grain size data for each thin section and derived quantities, including the eigenvalues and eigenvectors of the second-order orientation tensor. These data are available free of charge from the Australian Antarctic Data Centre (http://data.antarctica.gov.au) and are referenced by doi:10.4225/15/5669050CC1B3B.

*Acknowledgements.* The Australian Antarctic Division provided funding and logistical support for drilling the DSS ice core and subsequent data analysis through projects ASAC 15, AAS 757 and AAS 4289. The authors gratefully acknowledge the contribution of all participants in the Australian National Antarctic Research Expeditions associated with retrieval of the DSS ice core. Preparation of the data for archiving was supported by the Australian Government Cooperative Research Centres Programme through the Antarctic Climate and Ecosystems Cooperative Research Centre (ACE CRC). Discussions with J. L. Roberts assisted with data management and manuscript preparation. B. Raymond assisted with data control and hosting. AT thanks R.C. Warner for stressing the importance of making these data widely available to the glaciological community.





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
