# Peer review of "Ice crystal c-axis orientation and mean grain size measurements from the Dome Summit South ice core, Law Dome, East Antarctica"

_Earth System Science Data, 2015_

## Referee Comment (RC1) · M. Montagnat (Referee) · 20 Feb 2016

General comments:

The paper presents some data obtained along the Dome Summit South ice core, Antarctica, during the field seasons 1991-92 and 1992-93. These data consist of c-axis orientation measurements (textures) and mean grain size obtained on thin sections performed between 117 m depth and 1196 m depth. The measurements were performed in the field with the equipment available at that time, namely Rigsby stage for c-axis orientation data, and digitalised polaroid photographs for the grain size evaluation.

[Figure]

The DSS ice core site and its characteristics (mean temperature, accumulation, location...) are presented and the measurement techniques are detailed. Then the data are briefly shown on two figures and the data files that will be made available are detailed.

- Since the authors clearly explain the fact that there exist now modern tools to perform similar measurements (Automatic Ice Texture Analyser for instance) with a much higher spatial and angular resolution, we could expect more comments about the way to compare the presented data and the one that are nowadays measured. This could be important while willing to use the data in modern flow model that integrate the texture and its evolution within flow laws (Gagliardini et al. 2009, Bargmann, Seddik and Greeve 2011 for instance). In particular, the "old method" only enables to estimate 1 orientation per grain, and therefore does not account for the grain size effect on texture. It was shown by Gagliardini et al. 2004 (J. Glaciol. 50) that it can induce a noticeable effect on the texture evaluation. Such analysis of the limitation of the data set would greatly enhance the quality of this paper, and help people to know how they can use them. In particular, a comparison with some data measured with a modern analyser on a few characteristic sections (small grains, large grains areas) would help and enable to know if the DSS data obtained here can be used quantitatively or not.

- In the same idea, a large scattering of the c-axis orientation data is observed in figure 2. An evaluation of the error bar associated with the measurement would enable to see whether this scattering is due to the data itself (high accumulation zone, more impact of the layering of the snow mantle...), or to the limitation of the measurement technique and conditions. Durand et al. 2006 (J. Glaciol 52) techniques could probably be used to estimate a standard deviation of the second order tensor eigenvalues due to the limited number of grains. With modern AITA measurement, this limited number of grains effect was shown to be the main source of error by Montagnat et al. 2012 (EPSL).

- Concerning the grain size measurements, the authors could also comment on the fact that the technique they used can be considered, or not, close enough to modern techniques (based on the calculation of the number of pixels in each grain, via seg-
mentation) to enable direct comparison with recent measurements. Could the authors also comment on the impact of selecting 100 grains? Is there any bias, related to the fact that one will more easily chose the larger grains, is it operator dependant?

- Although we do not expect this paper to provide a deep scientific analysis, the authors could add in the figures 2 and 3 some climatic data, or at least locate the climatic transitions along the core, and give an idea of the dust content variation with depth? This could provide a first rough view of the interest a reader could have in using the data for further studies and comparison with other ice cores.

Specific comments:

- abstract line 5: "the" appears twice - p2 line 20: in most large scale ice sheet model, ice is considered as a viscoplastic material whose flow is modeled by the Glen flow law. And not a viscoelastic fluid! - p3 line 32: "orientationS" - p6 line 30: A1 and A2 are not defined - p6 line 7: maybe mention Russell-Head and Wilson 2001, J. Glaciol - p6 line 9: "the ability TO spatially map..." - p13 line 24: could you please show an example of the figure described?

Data files: the .mat matlab file is very easy to use and contains all the required information.

Since there exist only one DSS core, the data are unique and cannot be reproduced.

As a conclusion, I would recommend the publication of this paper since it provides data and the frame in which they where obtained, that could be of use for qualitative comparison between ice cores. It would nevertheless be important to answer the general comments mentioned before in order to enable a better understanding of the data, and therefore a clear idea of how they can be used, in particular in comparison with most recent data, or in the frame of ice flow modeling.

---

## Referee Comment (RC2) · Anonymous Referee #2 · 27 Mar 2016

This manuscript presents microstructural parameters of thin sections of the ice core recovered from the Law Ice Dome, Antarctica, in the 1990s. The data have been derived with state of the art methods of that time. To my knowledge these data have not been published before. Mean grain size of one thin section per recovered core meter are presented. The crystal fabric orientation was measured for about 100 grains per prepared thin section. The data are unique as the archived ice from the core is prioritized for chemical analysis, thus it is not possible to derive microstructural parameters with current methods, which allow for more detailed grain statistics. The authors give a detailed introduction to the data and its background, the location and the drilling campaign. They also describe in detail how the data could contribute to improve the

understanding of the rheological properties of ice, which is essential to improve flow modeling. The manuscript is very well structured and a pleasure to read. The data itself is easily accessible and for convenience provided in two different data formats. The pdf-files with the visualization of the fabric data, showing the stereographic plots and histograms, are a very useful addition, as they provide an immediate access to the information enclosed in the data. It is great that finally this valuable data set will be accessible for the glaciological community. It will surely be of use to many and should be published.

I would like to make two minor comments that the authors might want to consider before final publication.

1. In the discussion of the role of ice fabric in flow modeling the authors state that anisotropic flow models are currently not applied to the large scale. There are however some application of these models on the intermediate scale, which highlight the importance of anisotropy to the flow of ice (e.g. Zwinger et al. 2014) and the stratigraphy of ice divides (Martin et al., 2012). Maybe it would be appropriate to mention one of these here.

2. Grain sizes: Is there any information about the grain size distribution within the samples? Mean grain size is of course a valuable parameter but it would be great to have some idea about the shape and/or variation of size. Do the original polaroids still exist and could be digitized? Or could you maybe include some of them as figures in the paper?

References: Zwinger, et al., 2014, The Cryosphere, 8, 607–621, 2014, doi:10.5194/tc-8-607-2014 Martin and Gudmundsson, 2012, The Cryosphere, 6, 1221–1229, 2012, doi:10.5194/tc-6-1221-2012
* * *

---

## Author Comment (AC1) · 29 Apr 2016

**Comments to the Scientific Editor, Earth System Science Data Discussions**

**Manuscript: essd-2015-45**

**Ice crystal c-axis orientation and mean grain size measurements from the Dome Summit South ice core, Law Dome, East Antarctica**

**Authors: Adam TREVERROW, LI J., JACKA, T.H.**

**General comments to the editor:**

Dear Editor,

In the following we present our response to the two reviewers for manuscript essd-2015-45, *Ice crystal c-axis orientation and mean grain size measurements from the Dome Summit South ice core, Law Dome, East Antarctica* by Treverrow and others.

In the following, comments made by the reviewers have been reproduced in italic font.

**Response to reviewer 1, Maurine Montagnat (essd-2015-45-RC1)**

**R1 comment**: *These data consist of c-axis orientation measurements (textures) and mean grain size obtained on thin sections...*

**Authors response**: To avoid any misunderstanding some remarks on the usage of the terms 'fabric' and 'texture' are required since alternative definitions are occasionally used within the glaciological community, which can lead to ambiguities. Within the manuscript we use fabric to describe the pattern of c-axis orientations while texture refers to the size and shape characteristics of grains.

In some cases, fabric (as defined above) is considered to be a component of the overall texture. It is in this sense – where 'texture' is an umbrella term for microstructure which incorporates fabric – that reviewer 1 states '*These data consist of c- axis orientation measurements (textures)....*'. Similarly, this usage is implied in the acronym Automatic Ice Texture Analysers (AITA), used to describe the modern instruments for making c-axis orientation and other microstructural observations. We do not suggest either usage of the terms is incorrect and simply make note of the differences for clarity.

**R1 comment**: *Since the authors clearly explain the fact that there exist now modern tools to perform similar measurements (Automatic Ice Texture Analyser for instance) with a much higher spatial and angular resolution, we could expect more comments about the way to compare the presented data and the one that are nowadays measured. This could be important while willing to use the data in modern flow model that integrate the texture and its evolution within flow laws (Gagliardini et al. 2009, Bargmann, Seddik and Greeve 2011 for instance). In particular, the 'old method' only enables to estimate 1 orientation per grain, and therefore does not account for the grain size effect on texture. It was shown by Gagliardini et al. 2004 (J. Glaciol. 50) that it can induce a noticeable effect on the texture evaluation. Such analysis of the limitation of the data set would greatly enhance the quality of this paper, and help people to know how they can use them.*

**Authors response**: As noted by reviewer 1 the presented data contain a single c-axis orientation value for each grain, as opposed to the pixel-scale orientation data available from modern analytical instruments. As a result, each grain contributes equally to the orientation density and derived quantities, such as the second-order c-axis orientation tensor, $\Lambda$. We agree that the lack of grain-size weighted orientation data may, in some applications, impose restrictions on how the data are used. We have added further comments, based on Gagliardini and others (2004), explaining the potential effect of equal weighting of individual grain contributions to quantitative descriptions of microstructure.

**Original text, P12, L18–20**:

... As the area of individual grain orientations was not recorded, volume weighting of the orientation data (e.g. Durand and others, 2006) was not possible and all orientations contribute equally to $\Lambda$ (Fig. 2). ...

**Revised text, P12, L18 – P13, L3**:

... As the area of individual grain orientations was not recorded, volume weighting of the orientation data (e.g. Durand and others, 2006) was not possible and all orientations contribute equally to $\Lambda$ (Fig. 2).

Allowances can be made for the distribution of grain sizes encountered in polycrystalline materials by weighting the contribution of individual $c$-axis orientations according to their area in quantitative descriptions of fabric, such as $\Lambda$. The pixel-scale orientation data provided by modern automated fabric analysers makes area weighting of $c$-axis orientations a routine aspect of microstructural analysis. In turn, this allows for an improved representation of microstructures extracted from thin sections. For example, Gagliardini and others (2004) note that with area weighting of orientations, the mean error in second-order orientation tensor based representations of fabric (e.g. Woodcock, 1977; Durand and others, 2006) may be up to $\sim 2.5$ times less than for equal weighted orientations. Notwithstanding the restriction of this data set to equal weighting of orientations, it represents a valuable resource for the quantitative assessment of ice flow relations and microstructural evolution. ...

**R1 comment**: *In particular, a comparison with some data measured with a modern analyser on a few characteristic sections (small grains, large grains areas) would help and enable to know if the DSS data obtained here can be used quantitatively or not.*

**Authors comment**: A direct comparison of the presented DSS ice core $c$-axis orientation and mean grain area measurements with new measurements made with a modern automated instrument is not possible due to the small amount of remaining DSS core material. This is described in the original manuscript at:

**Original text, P6, L10–14**:

... Since analytical techniques for both microstructural and chemical analyses of ice cores are destructive, only a small proportion of the original ice core cross section remains over the full length of the core. This remainder is insufficient to allow a detailed reanalysis of c-axis orientations and grain size using a modern instrument. Additionally, the remaining core material is prioritised for chemical reanalyses, should any be required. ...

**Authors comment**: Regarding the quantitative application of the data, as noted above, the data represent a valuable resource to assist the development and/or evaluation of ice flow relations or microstructure evolution models – even with equal weighting of individual grain contributions to derived fabric parameters. As described by Durand and others (2006) new statistical descriptions of microstructures are made possible by modern instruments. Consequently some of the historically commonplace descriptions of fabric are no longer appropriate for modern data sets- and vice versa. Importantly, this does not mean the presented data can not used quantitatively.

**R1 comment**: *we could expect more comments about the way to compare the presented data and the one that are nowadays measured. This could be important while willing to use the data in modern flow model that integrate the texture and its evolution within flow laws (Gagliardini et al. 2009, Bargmann, Seddik and Greeve 2011 for instance),*

**Authors response**: We note that the data can, and have been, used quantitatively. In Treverrow and others (2015) the data were used in an evaluation of the anisotropic ice flow relations presented by Azuma and Goto-Azuma (1996); Thorsteinsson (2002); Seddik and others (2008) and Budd and

others (2013). In particular the flow relation presented by Seddik and others (2008) (described using the acronym CAFFE) is also the flow relation used in the regional ice sheet modelling studies of Seddik and others (2011) and Bargmann and others (2012). While the CAFFE flow relation can incorporate volume weighting of orientations, we demonstrate that it can be applied with equal weighting of orientations. Following the suggestion of reviewer 1 regarding the differences between equal and area-based weighting of individual grain contributions to quantitative descriptions of microstructure, we consider that the comments now included at P12, L18 – P13, L3 (see above) provide an adequate introduction to the restrictions of equal weighting. The analysis of Treverrow and others (2015) demonstrates that when using ice core $c$-axis orientation data to evaluate multiple anisotropic flow relations, differences in their formulation greatly overwhelms any uncertainty in the fabric data due to either natural variability or measurement methods.

**R1 comment**: *An evaluation of the error bar associated with the measurement would enable to see whether this scattering is due to the data itself (high accumulation zone, more impact of the layering of the snow mantle...), or to the limitation of the measurement technique and conditions. Durand et al. 2006 (J. Glaciol 52) techniques could probably be used to estimate a standard deviation of the second order tensor eigenvalues due to the limited number of grains. With modern AITA measurement, this limited number of grains effect was shown to be the main source of error by Montagnat et al. 2012 (EPSL).*

**Authors response**: Two factors contribute to the variability in the DSS ice core fabric measurements. First is the observed large-scale development of fabric from the surface to the bottom of the borehole, with a broad peak in the magnitude of the largest eigenvalue, $a_1$, of the orientation tensor, $\Lambda$, between depths of $\sim 800 - 1000\,\text{m}$. Second is variability, or noise in the data that is superimposed on the large-scale pattern of evolution with depth. We suggest that the small-scale variability in fabric evident over distances of several tens of metres results from a combination of natural variability and measurement methods.

While detailed commentary on the large-scale flow and microdynamic processes that contribute to the observed variability in fabric is beyond the scope of this manuscript we note that Morgan and others (1997) and Li and others (1998) indicate that both impurities and the dynamic conditions at the drilling site influence the microstructural evolution and hence local variability in fabrics and mean grain size. The effect of soluble and insoluble impurities on microstructure has been reported for other cores (e.g. Gow and Meese, 2007; Durand and others, 2007) and similar effects are no doubt at play here. Figure 8 from Morgan and others (1997) includes depth profiles of the DSS ice core mean grain area and mean crystal $c$-axis orientation measured from the 185 thin sections described in this paper. Morgan and others (1997) describe how over small vertical distances of several ten of metres variability in both the mean grain size and $c$-axis orientations exist. Furthermore their Figure 8 demonstrates that thin sections with stronger fabrics (lower mean $c$-axis colatitude) tend to be associated with smaller mean grain sizes when compared to adjacent thin sections with weaker fabrics (higher mean $c$-axis colatitude). Analysis of insoluble impurities in the DSS ice core by Li and others (1998) suggests that locally high levels of microparticles are related to maintaining a small mean grain size and reducing rates of fabric evolution due to pinning effects.

In order to quantify variability in fabric statistics reviewer 1 has suggested calculation of the standard deviation of the orientation tensor eigenvalues, following the method described by Durand and others (2006). They present a method of sub-sampling simulated orientation data that is also applicable to the high spatial resolution of data obtained from single thin sections using a modern automated fabric analyser. This high resolution data enables sub-sampling of the data in order to

estimate the component of variability associated with the number of orientations used to calculate the second-order orientation tensor and its eigenvalues. This technique is not appropriate for the DSS data since the limited number of $c$-axis orientations that can be routinely measured from thin sections using a Rigsby stage is too low for sub-sampling. However, based on the regression analysis of Durand and others (2006) we can use their Eqn. (26) to estimate the contribution of the population size (the sample size $N$) to the standard deviation, $\sigma_{a_i}^{\mathrm{P}}$, of the orientation tensor eigenvalues. Estimates of $\sigma_{a_i}^{\mathrm{P}}$ for each of the 185 thin sections are presented in a new Fig. 4 demonstrating how the standard deviation in orientation tensor values might be expected to vary according to the number, $N$, of $c$-axis orientations in each fabric. As expected, the maximum variability in $a_i$ associated with the smallest sample sizes (lowest $N$). This new figure and additional text (below) will improve the ability of those using the $c$-axis orientation data set to understand the level of variability and its sources.

**Revised text, P14, L4–P14,L9 & new figure**:

For a population of $N$ $c$-axis orientations the contribution of the population size to the standard deviation of the orientation tensor eigenvalues, $\sigma_{a_i}^{\mathrm{P}}$ can be estimated from,

$$\sigma_{a_i}^{\mathrm{P}} = \left[ -1.64 \times (a_1)^2 + 1.86 \times a_1 - 0.14 \right] \times \frac{1}{N^{1/2}}. \tag{3}$$

Equation (3) (Durand and others, 2006) was derived from the statistical analysis of multiple sub-samples of $100 < N < 1000$ orientations from a parent population of $10^4$ orientations. Values of $\sigma_{a_i}^{\mathrm{P}}$ calculated using Eqn. (3) for each of the 185 thin sections in the DSS data set are presented in Figure 4 and clearly demonstrate the influence of $N$ on the variability in fabric statistics; $\sigma_{a_i}^{\mathrm{P}}$ ranges from 0.0148 up to a maximum of 0.140 for the lowest values of N. As expected from Equation (3), $\sigma_{a_i}^{\mathrm{P}}$ decreases with larger $N$, and for the majority of the DSS data set, where $N > 40$ the estimated variability in $\sigma_{a_i}^{\mathrm{P}}$ is relatively low, as indicated by mean and median values of 0.0355 and 0.0337 respectively.

[Figure]

Figure 4: Variation of the standard deviation, $\sigma_{a_i}^{\mathrm{P}}$, of the eigenvalues of the second-order orientation tensor with $N$, the number of $c$-axis orientations measured in each of the 185 thin sections. Values of $\sigma_{a_i}^{\mathrm{P}}$ are calculated according to Eqn. (3). The median value of $N = 100$ and there are 50 measurements at $N = 100$ and 70 at $N = 102$.

A component of the observed variability in the DSS fabric and mean grain size data (Figs. 2 and 3) over small vertical distances (e.g. $\sim 5 - 20\,\mathrm{m}$) is due to the influence of ice chemistry, impurities and the dynamic conditions at the drilling site on microstructural evolution. This variability is therefore an inherent feature of the data and similar effects have been reported for other Antarctic and Greenland ice cores (e.g. Durand and others, 2007; Gow and Meese, 2007).

Estimating the magnitude of impurity effects on the variability in derived microstructural parameters is not possible with the DSS data set; however, observations from Morgan and others (1997, Fig. 8) indicate that localised variability in the fabric strength that is superimposed on the large-scale pattern of fabric evolution with depth, corresponds to changes in the mean grain size. In particular, thin sections with stronger fabrics (lower mean $c$-axis colatitude) tended to have smaller mean grain sizes than adjacent thin sections with weaker fabrics and correspondingly larger mean grain sizes. Based on the analysis of the insoluble impurity content in the DSS ice core, Li and others (1998, Fig. 2) suggest that locally high levels of microparticles are associated with a refinement of the mean grain size and the preservation of stronger crystal orientation fabrics due to a retardation of recrystallisation processes.

**R1 comment**: *Concerning the grain size measurements, the authors could also comment on the fact that the technique they used can be considered, or not, close enough to modern techniques (based on the calculation of the number of pixels in each grain, via segmentation) to enable direct comparison with recent measurements.*

**Authors comment**: It is clear the calculation of mean grain area based on the selection of up $\sim 100$ grains provides an inferior estimate of grain size in comparison to analyses by modern instruments from which individual grain size can be routinely extracted. One option for those wishing to directly compare these mean DSS grain sizes with measurements from other sites obtained using a modern instrument is to calculate mean grain areas from their higher resolution data. The drawbacks associated with this approach have been addressed above.

As the original DSS thin sections no longer exist reanalysis with a modern fabric analyser is not possible. For those wishing to investigate variations in DSS ice core grain size in greater detail a full set of polaroid photographs of thin sections obtained between orthogonal plane polarising filters remain in existence. These can be accessed by contacting the authors.

**Revised text, P8 L3-L9**: The single mean grain size measurements per thin section included in this data set provide a coarse representation of grain size in comparison with the data that can be obtained using modern instruments. Since the original thin sections used for $c$-axis and mean grain size measurements no longer exist, higher resolution analyses using such an instrument are not possible. For those interested in extracting additional microstructural information, such as the distribution of grain size and/or shape, digital analysis of the original thin section images is a possibility. A full set polaroid photographs of the DSS ice core thin sections remain in existence and these can be accessed by contacting the authors via the Australian Antarctic Data Centre (http://data.antarctica.gov.au).

**R1 comment**: *Could the authors also comment on the impact of selecting 100 grains? Is there any bias, related to the fact that one will more easily chose the larger grains, is it operator dependant?*

**Authors comment**: Bias towards the selection of larger grains for $c$-axis orientation measurement was avoided by selecting up to $\sim 100$ neighbouring grains within an enclosed region. This required tracking those grains which had been measured on a polaroid image of the thin section which is common practice when using a Rigsby stage as it avoids repeat measurements of the same grain.

**Revised text, P6 L5-L8**:

Sampling bias, including the preferential selection of larger grains was avoided by i) tracking those

grains that had been measured on a polaroid image of the thin section and ii) selecting a continuous (i.e. gap-free) set of neighbouring grains from within an enclosed region so that each grain had at least one neighbour within the region of interest.

**R1 comment**: *Although we do not expect this paper to provide a deep scientific analysis, the authors could add in the figures 2 and 3 some climatic data, or at least locate the climatic transitions along the core, and give an idea of the dust content variation with depth? This could provide a first rough view of the interest a reader could have in using the data for further studies and comparison with other ice cores.*

**Authors comment**: There are clearly pros and cons to the suggestion to add climate and/or ice core chemistry data to either of Figs. 2 and 3. As the reviewer suggests this would provide further context for the data; however what constitutes useful context will vary according to the specific end use of the data and we don't wish to preempt this.

Our preference is to not include additional climate and ice core chemistry in figures within this publication. Such data do not form a part of the data products which are described in this manuscript and made available at the Australian Antarctic Data Centre.

We have added references to Morgan and others (1997) and Li and others (1998) (see above for revised text at P14, L4–P14). These references include figures that may be of use to some users of the dataset. Li and others (1998) discuss the role of microparticle concentrations measured in the DSS ice core on microstructural development. Morgan and others (1997) describe the corresponding variability observed in the $c$-axis and grain size records. They also present $\delta^{18}O$ (‰) and depth-age data.

**Specific comments: Reviewer 1**

**abstract line 5**: *?the? appears twice* Typo corrected.

**p2 line 20**: *in most large scale ice sheet model, ice is considered as a viscoplastic material whose flow is modeled by the Glen flow law. And not a viscoelastic fluid!* Polycrystalline ice is viscoelastic material; however, the reviewer is correct in that most ice flow relations used in ice sheet models treat ice as a viscoplastic material. Therefore 'viscoplastic' is correct in this sentence. The suggested correction has been made.

**p3 line 32**: *?orientationS?* This is actually P2, L32. Orientation is correct here. No change made.

**p6 line 30**: *A1 and A2 are not defined* Reference to the A1 and A2 scales on the Rigsby stage is actually a distraction from the purpose of this sentence. A better solution is to remove reference to A1 and A2 scales altogether.

**Previous text**: Because the universal stage used to measure the DSS ice core crystal orientations employs sensors to determine the position of the instrument axes, incorrect reading of the A1 and A2 axis scales is eliminated as a source of error in these data.

**Revised text**: Because the universal stage used to measure the DSS ice core crystal orientations employs sensors to determine the position of the instrument axes, incorrect reading of the axis scales is eliminated as a source of error in these data.

**p6 line 7**: *maybe mention Russell-Head and Wilson 2001, J. Glaciol*

There does not appear to be a 2001 publication by Russell-Head & Wilson in the Journal of Glaciology. Perhaps the reviewer was referring to:

Russell-Head, D. S., and Wilson C. J. L. (2001), Automated fabric analyser system for quartz and ice, Abstr. Geol. Soc. Aust., 64, 159?

This abstract-only publication seems to be the only article from these authors in 2001. It describes a precursor to the Russell-Head Instruments G50 automated ice fabric analyser (the basic principles

of operation are similar to the G50). Since the G50 instrument is the most commonly used automated fabric analyser in modern glaciological laboratories it is best described by the references already included the manuscript. The references included in the manuscripts are also more easily accessible than the above abstract-only publication.

**p6 line 9**: *?the ability TO spatially map...* Missing word inserted.

**p13 line 24**: *could you please show an example of the figure described?* Figure 5 added.

**Response to reviewer 2, (anonymous, essd-2015-45-RC2)**

**R2 comment**: *In the discussion of the role of ice fabric in flow modeling the authors state that anisotropic flow models are currently not applied to the large scale. There are however some application of these models on the intermediate scale, which highlight the importance of anisotropy to the flow of ice (e.g. Zwinger et al. 2014) and the stratigraphy of ice divides (Martin et al., 2012). Maybe it would be appropriate to mention one of these here.*

**Authors comment**: The text has been updated to include comments on how regional-scale ice sheet models have been used to demonstrate the importance of including a description of anisotropic rheology to accurately model ice sheet dynamics.

**Revised text, P2, L18-L23**: In general such flow relations are not suited to implementation within models used to simulate the large-scale evolution of the polar ice sheets, being either too numerically complex or lacking the ability to accurately describe anisotropic flow effects (e.g. Treverrow and others, 2015). The importance of including a description of anisotropic ice rheology to accurate modelling of ice sheet dynamics has been demonstrated in regional-scale ice sheet models (e.g. Seddik and others, 2011; Zwinger and others, 2014) where the task of determining the three dimensional distribution of stresses within an ice mass is computationally tractable.

**R2 comment**: *Grain sizes: Is there any information about the grain size distribution within the samples? Mean grain size is of course a valuable parameter but it would be great to have some idea about the shape and/or variation of size. Do the original polaroids still exist and could be digitized? Or could you maybe include some of them as figures in the paper?*

**Authors comment**: At the time of drilling and thin section analysis no details of the grain size and shape distributions within individual thin sections were recorded.

Copies of the original polaroid images remain in existence. In preparing this data set we didn't digitize these polaroids as it is difficult to preempt what research use these may have and therefore what scan quality or file type is appropriate for a given application. In this case we feel it is best to simply include a statement on the existence of the polaroids and their availability for use.

*Revised text, P8, L6-L9*: For those interested in extracting additional microstructural information, such as the distribution of grain size and/or shape, digital analysis of the original thin section images is a possibility. A full set polaroid photographs of the DSS ice core thin sections remains in existence and these can be accessed by contacting the authors via the Australian Antarctic Data Centre (http://data.antarctica.gov.au).

Adam Treverrow
April 29, 2016

**References**

Azuma, N. and K. Goto-Azuma, 1996. An anisotropic flow law for ice sheet ice and its implications, *Annals of Glaciology*, **23**, 202–208.

Bargmann, S., H. Seddik and R. Greve, 2012. Computational modeling of flow-induced anisotropy of polar ice for the EDML deep drilling site, Antarctica: The effect of rotation recrystallization and grain boundary migration, *International Journal for Numerical and Analytical Methods in Geomechanics*, **36**(7), 892–917.

Budd, W.F., R.C. Warner, T.H Jacka, J. Li and A. Treverrow, 2013. Ice flow relations for stress and strain rate components from combined shear and compression laboratory experiments, *Journal of Glaciology*, **59**(214), 374–392.

Durand, G., O. Gagliardini, T. Thorsteinsson, A. Svensson, S. Kipfstuhl and D. Dahl-Jensen, 2006. Ice microstructure and fabric: an up-to-date approach for measuring textures, *Journal of Glaciology*, **52**(179), 619–630.

Durand, G., F. Gillet-Chaulet, A. Svensson, O. Gagliardini, S. Kipfstuhl, J. Meyssonnier, F. Parrenin, P. Duval and D. Dahl-Jensen, 2007. Change in ice rheology during climate variations – implications for ice flow modelling and dating of the EPICA Dome C core, *Climate of the Past*, (3), 155–167.

Gagliardini, O., G. Durand and Yun Wang, 2004. Grain area as a statistical weight for polycrystal constituents, *Journal of Glaciology*, **50**(168), 87–95.

Gow, A.J. and D.A. Meese, 2007. The distribution and timing of tephra deposition at Siple Dome, Antarctica: possible climatic and rheologic implications, *Journal of Glaciology*, **53**(183), 585–596.

Li, J., T.H. Jacka and V. Morgan, 1998. Crystal-size and microparticle record in the ice core from Dome Summit South, Law Dome, East Antarctica, *Annals of Glaciology*, **27**, 343–348.

Morgan, V.I., C.W. Wookey, J. Li, T.D. van Ommen, W. Skinner and M.F. Fitzpatrick, 1997. Site information and initial results from deep ice drilling on Law Dome, Antarctica, *Journal of Glaciology*, **43**(143), 3–10.

Seddik, H., R. Greve, L. Placidi, I. Hamann and O. Gagliardini, 2008. Application of a continuum-mechanical model for the flow of anisotropic polar ice to the EDML core, Antarctica, *Journal of Glaciology*, **54**(187), 631–642.

Seddik, H., R. Greve, T. Zwinger and L. Placidi, 2011. A full Stokes ice flow model for the vicinity of Dome Fuji, Antarctica, with induced anisotropy and fabric evolution, *The Cryosphere*, **5**(doi:10.5194/tc-5-495-2011), 495–508.

Thorsteinsson, T., 2002. Fabric development with nearest-neighbour interaction and dynamic recrystallization, *Journal of Geophysical Research, Solid Earth*, **107**(B1).

Treverrow, A., R.C. Warner, W.F. Budd, T.H. Jacka and J. L. Roberts, 2015. Modelled stress distributions at the Dome Summit South borehole, Law Dome, East Antarctica: a comparison of anisotropic ice flow relations, *Journal of Glaciology*, **61**(229), 987–1004.

Woodcock, N.H., 1977. Specification of fabric shapes using an eigenvalue method, *Geological Society of America Bulletin*, **88**(9), 1231–1236.

Zwinger, T., M. Schäfer, C. Martín and J. C. Moore, 2014. Influence of anisotropy on velocity and age distribution at Scharffenbergbotnen blue ice area, *The Cryosphere*, **8**(2), 607–621.